# ai-corona: Radiologist-assistant deep learning framework for COVID-19 diagnosis in chest CT scans

Mehdi Yousefzadeh[1,2,3☯], Parsa Esfahanian[1☯], Seyed Mohammad Sadegh Movahed[3], Saeid Gorgin[1,4], Dara Rahmati[1,5], Atefeh Abedini[6], Seyed Alireza Nadji[7], Sara Haseli[6], Mehrdad Bakhshayesh Karam[6], Arda Kiani[6], Meisam Hoseinyazdi[8], Jafar Roshandel[6], Reza Lashgari[2]*

1 School of Computer Science, Institute for Research in Fundamental Sciences (IPM), Tehran, Iran, 2 Institute of Medical Science and Technology, Shahid Beheshti University, Tehran, Iran, 3 Department of Physics, Shahid Beheshti University, Tehran, Iran, 4 Department of Electrical Engineering and Information Technology, Iranian Research Organization for Science and Technology (IROST), Tehran, Iran, 5 Department of Computer Science and Engineering, Shahid Beheshti University, Tehran, Iran, 6 Chronic Respiratory Diseases Research Center, National Research Institute of Tuberculosis and Lung Diseases (NRITLD), Shahid Beheshti University of Medical Sciences and Health Services, Tehran, Iran, 7 Virology Research Center, National Research Institute of Tuberculosis and Lung Diseases (NRITLD), Shahid Beheshti University of Medical Sciences and Health Services, Tehran, Iran, 8 Department of Radiology, Medical Imaging Research Center, Shiraz University of Medical Sciences, Shiraz, Iran

☯ These authors contributed equally to this work.
* rezalashgari@gmail.com

**Data Availability Statement:** Data cannot be shared publicly because of ethical restrictions. CT scan data of this research from the Masih Daneshvari Hospital, including images and results,

## Abstract

The development of medical assisting tools based on artificial intelligence advances is essential in the global fight against COVID-19 outbreak and the future of medical systems. In this study, we introduce ai-corona, a radiologist-assistant deep learning framework for COVID-19 infection diagnosis using chest CT scans. Our framework incorporates an EfficientNetB3-based feature extractor. We employed three datasets; the CC-CCII set, the MasihDaneshvari Hospital (MDH) cohort, and the MosMedData cohort. Overall, these datasets constitute 7184 scans from 5693 subjects and include the COVID-19, non-COVID abnormal (NCA), common pneumonia (CP), non-pneumonia, and Normal classes. We evaluate ai-corona on test sets from the CC-CCII set, MDH cohort, and the entirety of the MosMedData cohort, for which it gained AUC scores of 0.997, 0.989, and 0.954, respectively. Our results indicates ai-corona outperforms all the alternative models. Lastly, our framework's diagnosis capabilities were evaluated as assistant to several experts. Accordingly, We observed an increase in both speed and accuracy of expert diagnosis when incorporating ai-corona's assistance.

## Introduction

Since the beginning of 2020, the novel Coronavirus Disease 2019 (COVID-19) has widely spread globally. As of 2021/04/29 08:42:32, there are more than 118 million reported cases and

are anonymous and non-personally identifiable, due to sensitive human study participant data. Data are available from the Institutional Data Access / Ethics Committee of Masih Daneshvari Hospital (contact via dr.abedini110@sbmu.ac.ir) for researchers who meet the criteria for access to confidential data. The MDH cohort is also available upon request to the corresponding author's email address. The CC-CCII set and the MosMedData cohort are publicly available at http://ncov-ai.big.ac.cn/download and https://mosmed.ai/en/, respectively.

**Funding:** The authors received no specific funding for this work.

**Competing interests:** The authors have declared that no competing interests exist.

2.5 million deaths [1]. Patients infected with COVID-19 commonly display symptoms such as fever, cough, fatigue, breathing difficulties, and muscle ache [2–4]. Vaccination has started in many countries since early 2021, but has been facing many challenges [5].

Currently, the most common method of testing for COVID-19 is Real-Time Polymerase Chain Reaction (RT-PCR) to detect viral nucleotides from upper respiratory specimens obtained by nasopharyngeal, oropharyngeal, or nasal mid-turbinate swab [6]. It has been shown that RT-PCR has several drawbacks. Reports suggest that since oropharyngeal swabs tend to detect COVID-19 less frequently than nasopharyngeal swabs, RT-PCR tends to have a high false-negative rate. Furthermore, RT-PCR has demonstrated a decrease in sensitivity to below 70% due to a low viral nucleic acid load and inefficiencies in its detection. This might be caused by immature development of nucleic acid detection technology, variation in detection rate by using different gene region targets, or a low patient viral load [7]. Besides, the availability of test kits and expert personnel to take them is still suboptimal in some countries. Not to mention the extended time period for the test completion contributes to ruling out RT-PCR as a reliable early detection and screening method [8–10]. In contrast to RT-PCR, diagnosis from other measurements such as chest Computed Tomography (CT) and blood factors is shown to be an effective early detection and screening method with high sensitivity in both detection [11] and anticipation of the severity of the disease [12].

Chest CT scan of a COVID-19 infected patient reveals bilateral peripheral involvement in multiple lobes with areas of consolidation and ground-glass opacity that progresses to "crazy-paving" patterns as the disease develops [11]. Asymmetric bilateral subpleural patchy ground-glass opacities and consolidation with a peripheral or posterior distribution, mainly in middle and lower lobes, are described as the most common image finding of COVID-19 [13]. To elaborate more, additional common findings include interlobular septal thickening, air bronchogram, and crazy paving pattern in the intermediate stages of the disease [11]. The most common pattern in the advanced stage is subpleural parenchymal bands, fibrous stripes, and subpleural resolution. Nodules, cystic change, pleural effusion, pericardial effusion, lymphadenopathy, cavitation, CT halo sign, and pneumothorax are some of the uncommon but possible findings [11, 14]. Recent studies indicate that organizing pneumonia, which occurs in the course of viral infection, is pathologically responsible for the clinical and radiological manifestation of Coronavirus pneumonia [13].

Deep learning is an area of Artificial Intelligence (AI) that has demonstrated tremendous capabilities in image feature extraction and has been recognized as a successful tool in medical imaging-based diagnosis, performing exceptionally with modalities such as X-Ray, Magnetic Resonance Imaging (MRI), and CT [15–21]. Recently, the research of AI-assisted respiratory diagnosis, especially pneumonia, has gained a lot of attention. One of the well-established standards in this research is the comparison of AI with expert medical and radiology professionals. As a pioneering work in this field [22], introduced a radiologist-level deep learning framework trained and validated on the ChestX-ray8 dataset [23] for the detection of 14 abnormalities, including pneumonia, in chest X-Ray images, which was further developed to propose a deep learning framework with pneumonia detection capabilities equivalent to that of expert radiologists [24]. Moreover [25], introduced a novel dataset of chest X-Ray images annotated with 14 abnormalities (7 the same as ChestX-ray8) and a state-of-the-art deep learning framework. Lastly [26], proposed a deep learning framework with a feature extractor based on AlexNet [27] to create a model capable of accurately diagnosing knee injuries from MRI scans and further showcases the positive impact of AI assistance in expert diagnosis.

In COVID-19 related research [8], has reported a sensitivity of 0.59 for RT-PCR test kit and 0.88 for CT-based diagnosis for patients with COVID-19 infection, and a radiologist sensitivity of 0.97 in diagnosing COVID-19 infected patients with RT-PCR confirmation. Furthermore

[28], introduces a deep learning framework with a 0.96 AUC score in the diagnosis of RT-PCR confirmed COVID-19 infected patients. Zhang *et al.*[29] proposed a model that on a dataset of 4154 subjects achieved an AUC score of 0.98 for diagnosing COVID-19 from two other classes; Normal and CP (Common Pneumonia *i.e.* non COVID-19 viral and bacterial pneumonia). They further made their dataset, CC-CCII [29], publicly available. In addition, the model proposed by Jin *et al.*[30], developed on a dataset of 9025 subjects, which is an amalgamation of their own data and several other public datasets (*e.g.* LIDC–IDRI [31], Tianchi-Alibaba [32], MosMedData [33], and CC-CCII), gained an accuracy of 0.975 for diagnosing between COVID-19 and three other classes (non-pneumonia, non-viral community-acquired pneumonia, Influenza-A/B), 0.921 for between COVID-19 and the CP and Normal classes on the CC-CCII dataset, and 0.933 for between COVID-19 from non-pneumonia on the MosMed-Data cohort. Further, this work manages to astoundingly diagnose between COVID-19 and influenza type-A, which is surprising given the small amount of influenza data in their study.

In this paper, we present *ai-corona*, a radiologist-level deep learning framework for COVID-19 diagnosis in chest CT scans. Our framework was developed on a set of 7184 lung CT scans from 5693 subjects, for which 2032 subjects are from the Masih Daneshvari Hospital (MDH) cohort and the rest belong to the CC-CCII set and MosMedData cohort. This data was gathered from three countries; China, Iran, and Russia. In this work, our framework diagnoses between COVID-19 and CP (common pneumonia), NCA (non COVID-19 abnormal), non-pneumonia, and Normal classes. We evaluate and compare the performance of *ai-corona* with experts and RT-PCR in COVID-19 diagnosis and further compare our framework with AI models proposed by Zhang *et al.* [29] and Jin *et al.* [30]. Finally, we examine the impact of AI as assistance to expert diagnosis.

In short, the main advantages and novelties of this study are as follows:

- Introducing a comprehensive and authentic methodology for annotating the dataset cases for such work, especially the COVID-19 infection, for the MDH dataset.

- Proposing a deep learning framework that is capable of accurately diagnosing chest CT scans for COVID-19, while being robust to the number of slices in the scan and having a low computational load.

- Thorough evaluation of the diagnosis performance of *ai-corona* on multiple datasets and comparing to radiologists, RT-PRC, and two other similar works.

- Evaluating and elucidating the impact of *ai-corona*'s assistance on radiologists' diagnosis performance.

## Materials and methods

### Data

Three datasets were employed in this work; The MDH cohort, the CC-CCII set, and the Mos-MedData cohort. An overall summery of all the data employed in our work can be found in Table 1.

The first dataset was obtained by our group from patients hospitalized at the Masih Daneshvari Hospital (MDH) (Tehran, Iran). The cascade structure of this cohort can be found in S1 Fig. This cohort consists of 2121 lung CT scans from 2032 subjects annotated into 3 classes: (1) Normal; (2) Non-COVID Abnormal (NCA); and (3) COVID-19. Since differentiating between COVID-19 and Normal classes is easier than between COVID-19 and NCA (especially if there are similar imaging features), having the NCA class is very important, as it

**Table 1. Number of subjects and (number of scans) for each class in the CC-CCII set, MDH cohort, and MosMedData cohort separated over training, tuning, and test.**

| Dataset | Classes | Training | Tuning | Test | # |
|---|---|---|---|---|---|
| MDH | Normal | 467 (470) | 51 (51) | 120 (121) | 628 (642) |
| | NCA | 576 (578) | 64 (64) | 117 (117) | 757 (759) |
| | COVID-19 | 475 (542) | 53 (59) | 109 (119) | 637 (720) |
| CC-CCII | Normal | 670 (844) | 72 (94) | 81 (105) | 823 (1043) |
| | CP | 665 (1131) | 67 (127) | 87 (147) | 828 (1405) |
| | COVID-19 | 734 (1231) | 82 (131) | 84 (143) | 900 (1505) |
| MosMedData | Non-Pneumonia | - | - | 254 (254) | 254 (254) |
| | COVID-19 | | | 856 (856) | 856 (856) |
| # | | 3587 (4796) | 398 (526) | 1708 (1862) | 5693 (7184) |

includes abnormalities such as atelectasis, cardiomegaly, lung emphysematous, hydropneumothorax, pneumothorax, cardiopulmonary edema, cavity, fibrocavitary changes, fibrobronchiectatic, mass, and nodule. Using the search function of the hospital's PACS and by reviewing reports by two board-certified radiologists, we gathered a preliminary dataset with a balanced distribution over all three classes.

All the participants in the MDH cohort gave written consent and our work has received the ethical license of IR.SBMU.NRITLD.REC.1399.024 from the Iranian National Committee for Ethics in Biomedical Research.

Cases in the Normal and NCA classes are from prior to the start of the Coronavirus global pandemic. A subset of the data in these two classes was randomly selected for testing. This portion was re-annotated by a different expert radiologist. Only the cases with consistent labels (*i.e.* same label as in the initial report) were retained in the test set. The MDH Normal and NCA cases that were not included in the test subset were further divided randomly into a training subset and a tuning subset.

The MDH COVID-19 group scans for testing were taken in the early stages of the infection and included 119 lung CT scans from 109 patients hospitalized for more than three days. These scans were selected by the consensus of several metrics that indicate COVID-19 infection: (1) report by at least one radiologist on the scan; (2) confirmation of infection by two pulmonologists; (3) clinical presentation; and (4) RT-PCR report.

Furthermore, unlike other works that take a positive RT-PCT as the sole criterion to annotate a case with COVID-19 label, and since our evaluation includes comparing the diagnosis performance of *ai-corona* with experts and RT-PCT, we clearly could not use a dataset that was annotated solely based on RT-PCR test result. Our annotation strategy is, therefore, more comprehensive and incorporates additional available metadata.

The MDH COVID-19 training (1518 subjects, 1590 scans) and tuning (168 subjects, 174 scans) sets were annotated using the aforementioned reports by the two radiologists.

The CT scans in the MDH cohort contained between 21 to 46 slices acquired in axial orientation with a slice thickness between 8 and 10 mm, The histogram representation for the number of slices is indicated in S2(a) Fig, while S2(b) and S2(C) Fig illustrate the age and sex distribution of the MDH cohort.

Moreover, as the NCA class of the MDH cohort includes many samples with non COVID-19 pneumonia, we can take this class as the equivalent of the CC-CCI set CP class for our model's training.

The second dataset employed in this work was the publicly available CC-CCII dataset [29]. After quality control (*e.g.* removing non-standard scans such as those with a small number of

slices), this set contains 3953 CT scans from 2551 subjects. The scans in CC-CCII are anno-tated into three classes: Normal, Common Pneumonia (CP), and COVID-19. This CC-CCII dataset was randomly split into three subsets for: (1) training (2069 subjects, 3206 scans), (2) tuning (230 subjects, 352 scans), and (3) testing (252 subjects, 395 scans). The tuning subset was used for model checkpoint and selection of the best overall model.

The third dataset, MosMedData cohort, is also publicly available and is comprised of 1110 CT scans from 1110 subjects. This dataset is annotated into two classes: Non-pneumonia and COVID-19. We used the entire MosMedData cohort for *external* testing, that is, testing on a dataset that has not been used for model training or tuning. To evaluate our model on this cohort, we take the prediction of the COVID-19 class (for binary classification).

The public datasets LIDC–IDRI31 and Tianchi-Alibaba32 (which were used for the training of the model proposed by Jin *et al.* [30]) were not used in our framework's development, as these sets are for benign and malignant tumor diagnosis and they might introduce uncertain-ties to our framework.

For the RT-PCR evaluation set, 2672 subjects, each hospitalized for more than three days, were tested 6419 times between February to October 2020. Respiratory samples including pha-ryngeal swabs/washing were obtained from the subjects. Nucleic acid was extracted from the samples using a QiaSymphony system (QIAGEN, Hilden, Germany) and SARS-CoV-2 RNA was detected using primer and probe sequences for screening and conformation on the basis of the sequence described by [34]. An RT-PCR diagnosis is considered correct when a patient has at least one positive test result.

This project has received the ethical license of IR.SBMU.NRITLD.REC.1399.024 from the Iranian National Committee for Ethics in Biomedical Research.

## Pre-processing

For all the image slices, the top 0.5% of pixels with the highest values were selected and their values were clipped to the lowest one in the range. Then, the intensities were linearly trans-formed to the range [0, 255]. Since we utilize models pre-trained on the ImageNet dataset [35], an additional ImageNet normalization was also carried out.

We also opted to not perform any segmentation (*i.e.* patch extraction) in our pre-process-ing. This is due to the manual annotation of each dataset (like Jin *et al.* [30]) being time and resource consuming. On the other hand, using automated methods, such as image processing techniques and pre-trained segmentation deep learning models, would introduce further unwanted error and uncertainty to our data, and subsequently, to the model's inference.

## Deep learning method

Inspired by [26], *ai-corona*'s deep learning model consists of two main blocks; a feature extrac-tor and a classifier. This is shown in Fig 1. The main challenge is mapping a 3-dimensional CT scan, which is a series of image slices, to a probability vector with a length equal to the number of classes. Another challenge is that all the scans not having the same number of slices and not all the slices being useful for diagnosis. To address this, we take the middle 50% image slices in each scan and denote the number of selected slices from each scan with $S$. We also experi-mented with other slice selection strategies (*e.g.* portion larger than 50%, top/bottom 50%, *etc.*), from which none performed better.

As shown in Fig 1, the feature extractor block is a pipeline, receiving each slice with dimen-sions $512 \times 512 \times 3$ (3 represents the number of color channels, but with all channels being exactly the same as for each image) and outputting a vector of length 1536 through an average pooling function. After all the slices have passed through the feature extractor block, we end

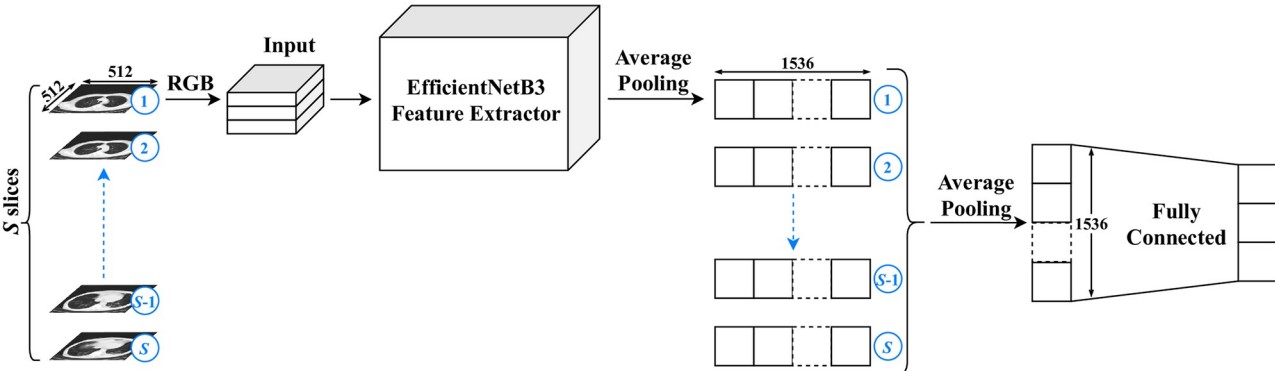

**Fig 1. The schematic structure of ai-corona's deep learning model.** The total number of utilized slices is labeled by S. Each selected slice is fed to the feature extractor block pipeline one by one so that we end up with S vectors, which are then transformed to a single vector via an average pooling function. Afterwards, the result is passed through a fully connected network to reach the three output neurons, corresponding to our three classes.

up with *S* vectors. After all the *S* slices have passed through the feature extractor block, another average pooling is applied to the results which yields a single vector of length 1536.

This pipeline manner ensures that our framework is independent of the number of slices in a CT scan, as we always end up with a single vector of length 1536 at the end of the feature extractor block. The pipeline receives different number of slices, extracts their features, and finally outputs a single vector of known length. Moreover, the use of only a single feature extractor significantly reduces the computational load of our framework, resulting in a much faster training and prediction time.

Convolutional Neural Networks (CNN) were used for the feature extraction block. We experimented with different CNN models, such as DenseNet, ResNet, Xception, and Efficient-NetB0 through EfficientNetB5 [36–39], taking into account their accuracy and accuracy density on the ImageNet dataset [40]. All of these models were initialized with their respective pre-trained weights on the ImageNet dataset. In the end, the EfficientNetB3 model stripped of its last dense layers was chosen as the primary feature extractor for our deep learning framework. The vector output of the EfficientNetB3 feature extraction block is then passed through the classifier block, which contains yet another average pooling layer that is connected to the model's output neurons corresponding to the classes via a dense network of connections.*ai-corona* is implemented with Python 3.7 [41] and Keras 2.3 [42] framework and was trained on NVIDIA GeForce RTX 2080 Ti for 60 epochs in a total of three hours. The Pydicom [43] package was used to read the DICOM file of the cases.

## Class activation maps

To generate the class activation map of an image slice, we computed a weighted average across the 1536 values of the feature vector using weights from the classification block to obtain a $10 \times 10$ image. The resulting map was then mapped to a color scheme, upsampled to $512 \times 512$ pixels, and overlaid with the original input image slice. Employing parameters from the classification block to weigh the feature vectors makes, more predictive features appear more bright. This leads to regions of the image slice that most influence the model's prediction to appear brighter. The class activation maps highlight which pixels in an image slice are important for the model's prediction [44].

## Statistical inference

In order to quantify the reliability of our findings and the performance of our results based on the model's detection of COVID-19 in chest CT scans, we provide a thorough comparison with expert practicing radiologists' diagnosis. To achieve a more conservative discrimination strategy, we compute the following evaluation criteria ranging from sensitivity (true positive rate), specificity (true negative rate), F1-score, Cohen's kappa, and finally to AUC. Moreover, the confusion matrix for all the classes of each individual study is also calculated.

We set the presence of the underlying class with a positive label and the rest of the classes assigned by a negative label. Incorporating error propagation and using the Bayesian statistics, we calculate the marginalized confidence region at a 95% level for each computed quantity. The significance of diagnostic results is examined by computing the *p*-value statistics systematically. To achieve a conservative decision, the $3\sigma$ significance level is usually considered.

Since the radiologists' diagnosis is given by "Yes" or "No" statements for each class, it is necessary to convert the probability values computed by our model to binary values. Hence, we selected an operating point for distinguishing a given case among others and compute the true positive rate (sensitivity) versus false positive rate (1-specificity). This operating point was selected such that the model would yield a high specificity. To make more sense, as well as the other mentioned evaluation criteria, the Receiver Operating Characteristic (ROC) diagram is also estimated for our studies. All of our criteria were calculated using the scikit-learn [45] package.

## Experts evaluation

Our team of experts annotated cases in the CC-CCII test set and MDH test set, with "Yes" and "No" labels for each class. To prevent a loss in experts' diagnosis performance due to fatigue, they were asked to work on small time chunks. Their performance was then evaluated and recorded. Next, to evaluate the impact of AI assistance on the experts' performance, after an appropriate amount of time and shuffling the sets (to prevent any remembrance), the experts re-annotated the two sets for a second time, while this time having access to the output of the model. They incorporated the model's opinion for suspicious cases on their own authority. Their performance was evaluated and recorded again.

Our team of four experts incorporates two practicing academic senior radiologists with 15 years of experience each. In our study, they're referred to as Senior Radiologist 1 and Senior Radiologist 2. Another expert is a practicing academic radiologist with 5 years of experience, which is referred to as Junior Radiologist. The last member is a senior radiology resident, referred to as Radiology Resident. The team of experts was chosen such that a wide range of experience and background knowledge would be present for our studies, in order to make it more comprehensive.

## Results

### Training, evaluation, and testing datasets

To develop *ai-corona*, we utilized data from three different sources: (1) the MDH cohort, (2) the publicly available CC-CCII dataset [29], and (3) the publicly available MosMedData cohort. The combined data were from multiple international sites and comprised of 7184 CT scans from 5693 subjects categorized into five classes: Normal, CP, NCA, non-pneumonia, and COVID-19. For a better comparison of the diagnosis performance between RT-PCR and CT scans, the RT-PCR test records of 2672 patients in a 7-month period were gathered.

The MDH and the CC-CCII data were used for training, evaluation (tuning), and testing. The MosMedData was used entirely for testing. Overall, 5322 scans from 3985 subjects were used for training and tuning, and three sets were used for testing: (1) CC-CCII test set (105 Normal, 147 CP, and 143 COVID-19 scans), (2) MDH test set (121 Normal, 117 NCA, and 119 COVID-19 scans), and (3) the entire MosMedData cohort (254 non-pneumonia and 856 COVID-19 scans).

Taking into consideration the ground truth annotation of all the works involved, the CC-CCII test set was used to compare *ai-corona* with the models proposed by Zhang *et al.* [29], Jin *et al.* [30], and with expert radiologists. Furthermore, the MDH test set was used to compare *ai-corona* with the radiologists and RT-PCR. Lastly, The MosMedData cohort was used to compare *ai-corona* with the model proposed by Jin *et al.* [30].

## RT-PCR sensitivity

Since the truth annotation methodology described in the Data subsection yields accurate labels, it was used to annotate a separate set for RT-PCR evaluation. This set is used to showcase the evolution of RT-PCR's sensitivity over a period of 7 months in Fig 2 (sensitivity of each day is calculated as the average sensitivity of a 15-day period centered around that day). RT-PCR's sensitivity oscillates in the range [0.351, 0.722]. The decrease in sensitivity to 0.351 on April 29, 2020, is due to changing the specimen obtaining method to oropharyeal wash [46]. This changed later and nasopharyngeal and oropharyngeal swabs were used. The biggest value for RT-PCR's sensitivity in this evaluation is considered its best, denoted by *RT-PCR Best*.

## Performance evaluation and comparison

Having three test sets, our framework's COVID-19 diagnosis performance for the CC-CCII test set, MDH test set, and the MosMedData cohort for all the studies is evaluated (an operating point was selected for each study). The confusion matrices for our evaluation results can be found in Fig 3. Moreover, for the COVID-19 class, ROC curves are showcased in Fig 4 and a more thorough look using the four metrics is depicted in Fig 5a and 5b. At last, the complete numerical reports for this evaluation can be found in Table 2. Values denoted with "-" in the table indicate a lack of report.

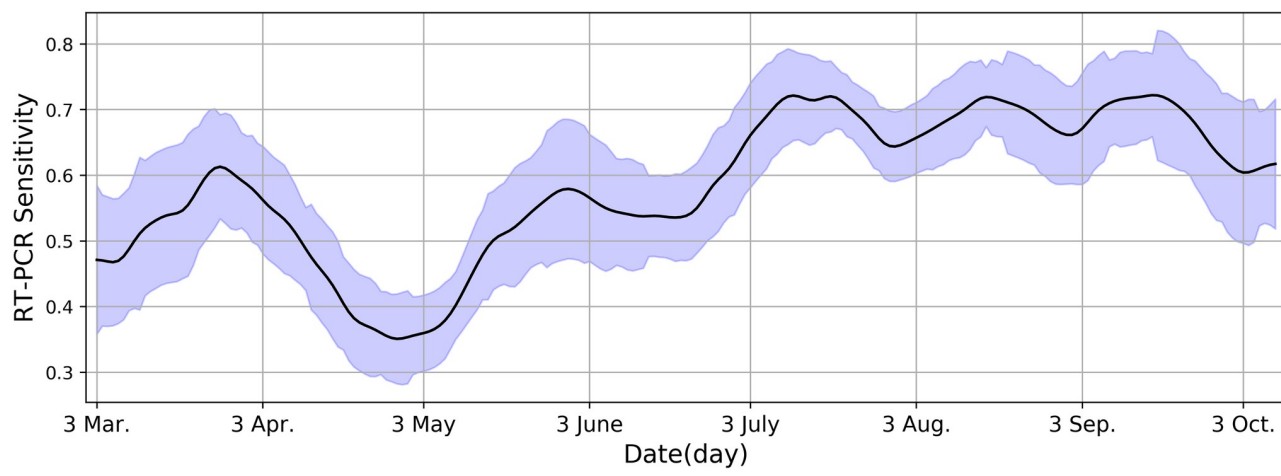

**Fig 2. Fluctuations in RT-PCR sensitivity on a daily basis.** The highest peak reaches 0.722, which is denoted as *RT-PCR Best*.

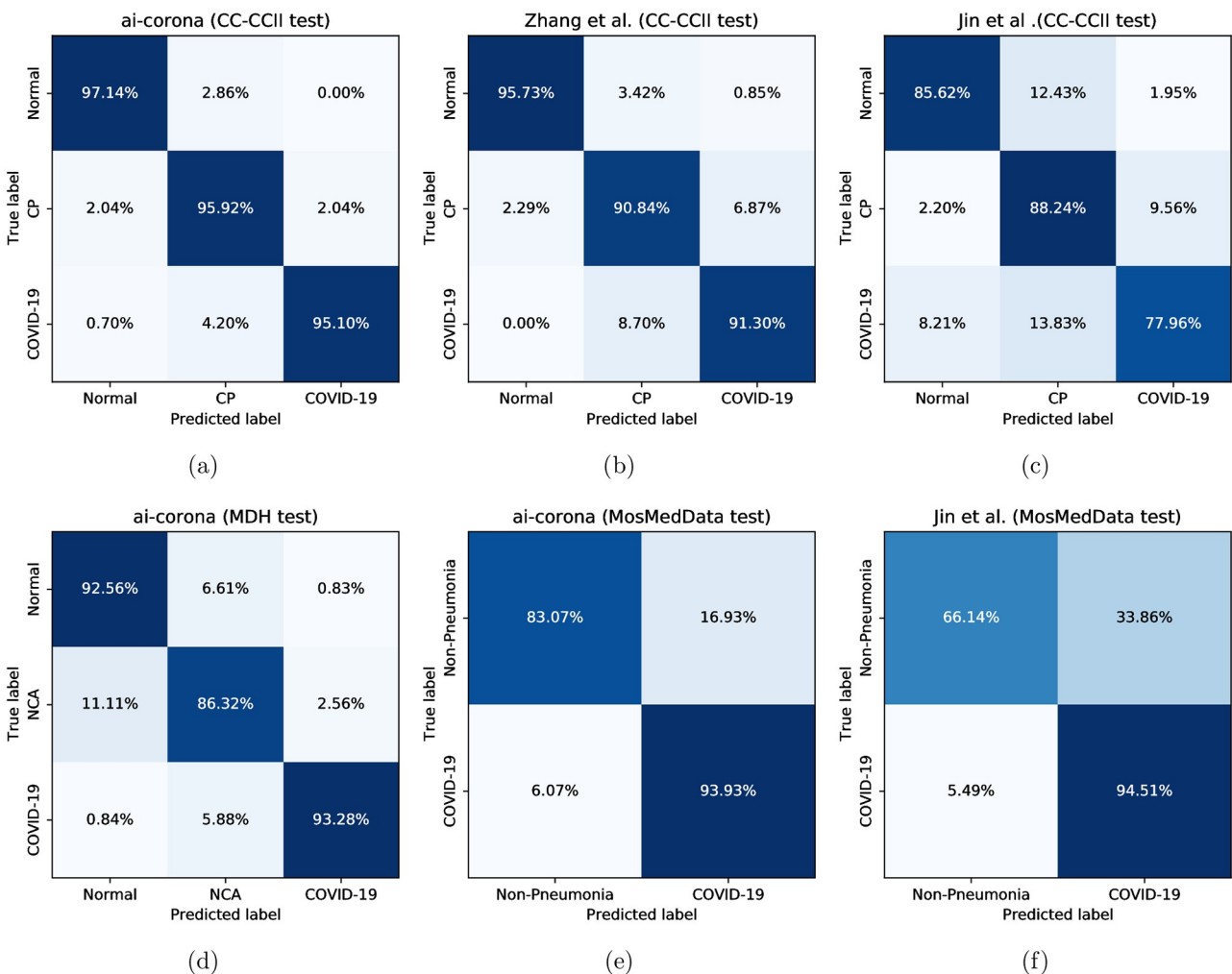

**Fig 3. Top row left-to-right: Confusion matrices for *ai-corona*, the model proposed by Zhang *et al.* [29], and the model proposed by Jin *et al.* [30] for the CC-CCI test set, respectively.** Bottom row left and middle: confusion matrices for *ai-corona* on the MosMedData cohort, respectively. Bottom row right: confusion matrix for the model proposed by Jin *et al.* [30] for the MosMedData cohort.

Fig 3a through Fig 3c show that *ai-corona* has performed better in all three classes (Normal, CP, COVID-19) compared to Zhang *et al.* [29] and Jin *et al.* [30] on the CC-CCII test set and achieves an AUC score of 0.997, sensitivity of 0.972, and specificity of 0.968 on the COVID-19 class. The confusion matrix in Fig 3d showcases our framework's performance on the MDH test set for the three classes of Normal, NCA, and COVID-19. For this dataset, our framework gains scores of 0.989, 0.924, and 0.983 for AUC, sensitivity, and specificity, respectively. In addition, Fig 3e and 3f showcase that our framework surpasses that of proposed by Jin *et al.* [30] on the MosMedData cohort with an AUC of 0.954. Although both have similar sensitivities in COVID-19 diagnosis, *ai-corona* outperforms Jin *et al.*'s model in non-pneumonia diagnosis with 83.07% accuracy, reporting fewer false positives.

The better diagnosis performance over the CC-CCII test set indicates that the task of diagnosing NCA from the other classes is indeed more difficult than diagnosing CP from the other classes. This due to all the different abnormalities present in the NCA class having their unique imaging features.

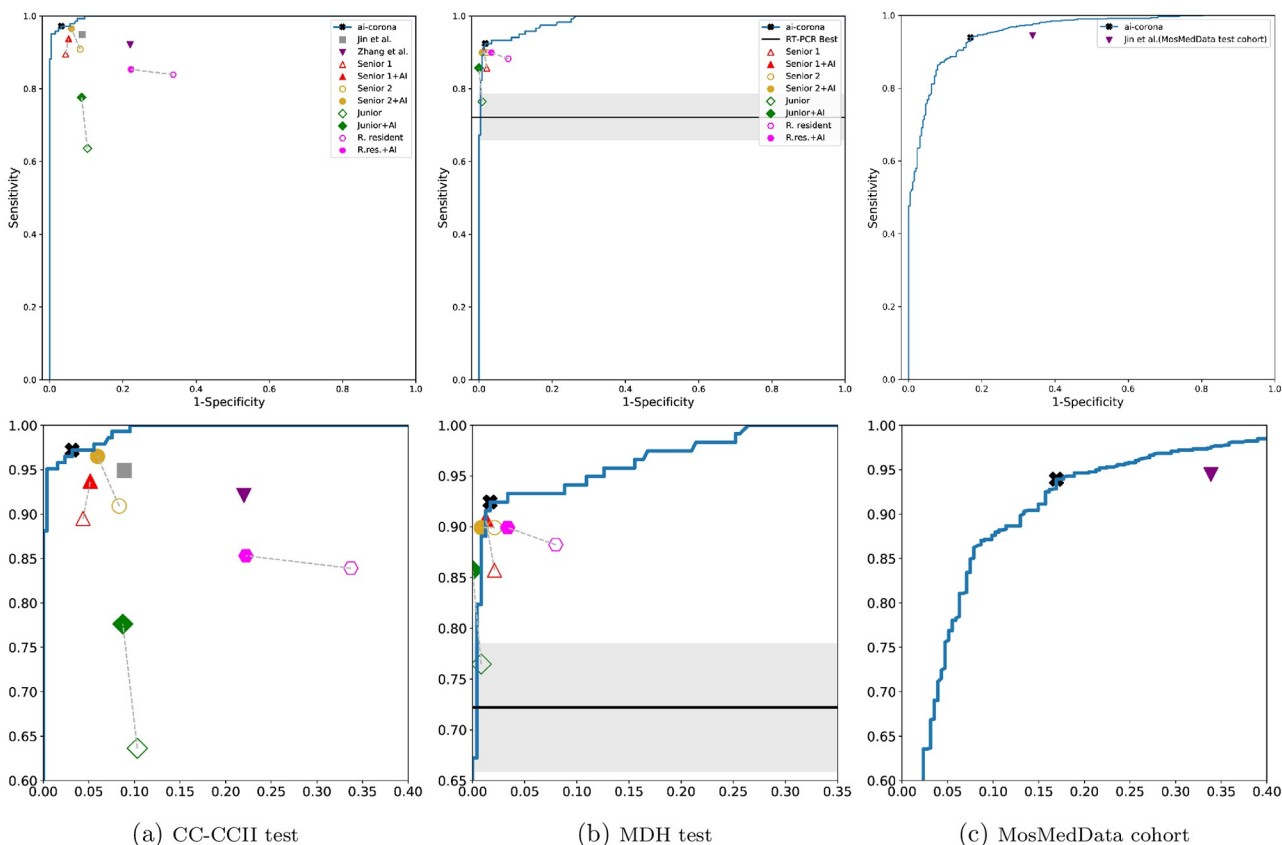

**Fig 4. ROC curve diagrams for *ai-corona* on the (a) CC-CCII test set (b) MDH test set (c) MosMedData cohort.** Diagrams in the bottom row correspond to a zoom-in of their respective curves. Hollow shapes represent an expert un-aided by AI, where filled shapes represent expert with AI assistance. As RT-PCR sensitivity was not available, its sensitivity is shown as a solid line in (b).

## Comparison with experts and RT-PCR

Fig 4(a) and the top diagram of Fig 5 showcase the COVID-19 diagnosis performance of *ai-corona* and its comparison with that of experts for the CC-CCII test set. As shown, our framework performs better in all cases (except for the specificity of Senior radiologist 1). Furthermore, Fig 4(a) and the bottom diagram of Fig 5 showcase the same comparison, but for the MDH test set. This time, the framework performed similar to radiologists in specificity, but outperformed in the other metrics. In this comparison, 93.3% of COVID-19 cases in the MDH test set (111 of 119) were diagnosed as infected by at least one expert. Out of the other 8 that were not, our framework managed to report one and RT-PCR reported three as infected. If RT-PCR was the only criteria for the truth annotation, the overall sensitivity of radiologists would improve to 97%, which would further confirm the findings in [8]. The complete reports for these two evaluations are in sections **a** and **b** of Table 2.

In Fig 4(b), the sensitivity of RT-PCR based diagnosis and CT based diagnosis is compared. The figure shows that *RT-PCR Best* sensitivity of 0.722 is lower than every expert diagnosing via CT. The *RT-PCR Best* sensitivity is an upper bound. Because if instead of testing patients hospitalized for more than three days, every COVID-19 admitted patient was tested, RT-PCR's sensitivity would be much lower than 0.722.

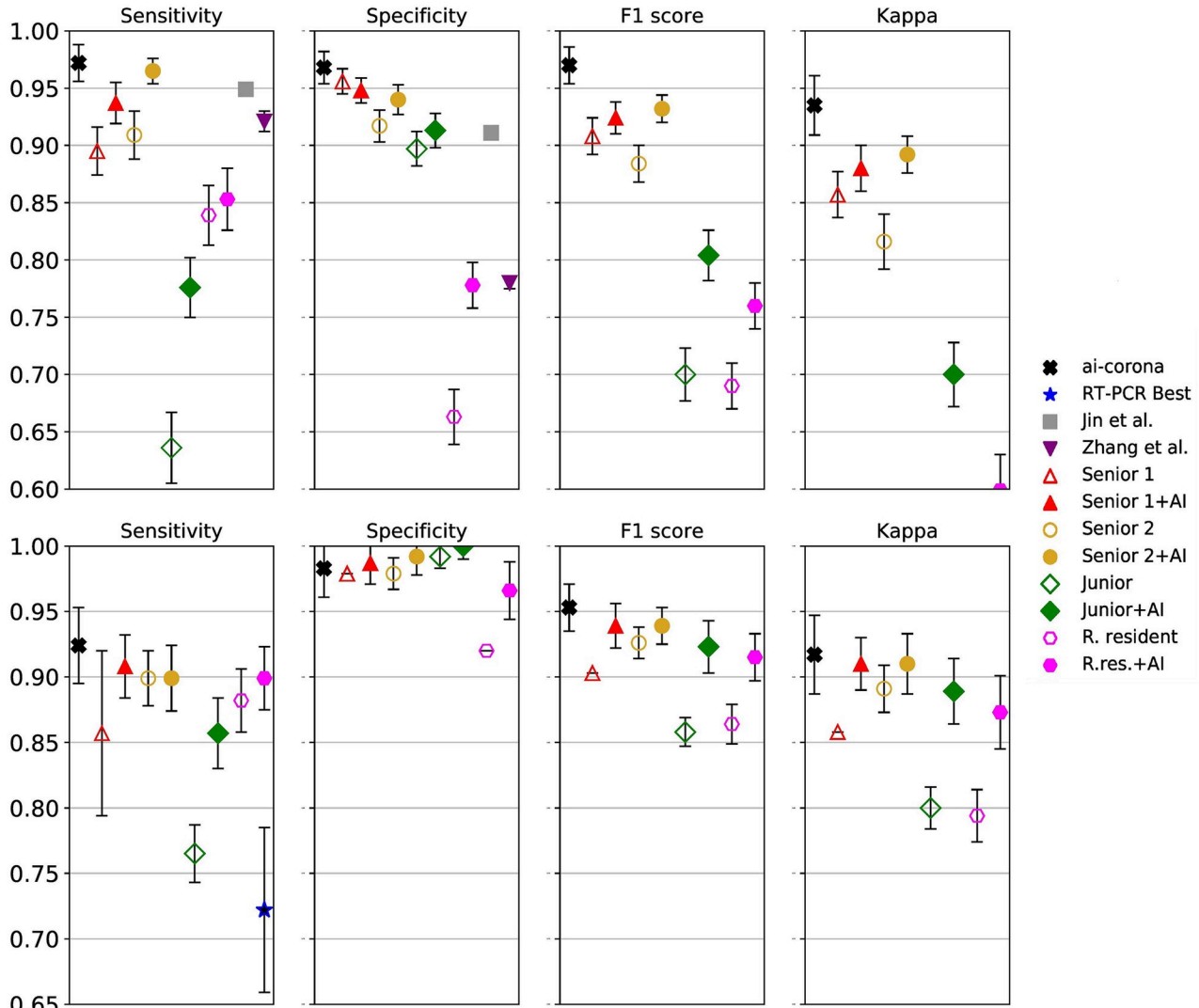

**Fig 5. Detailed comparison of all the studies using our evaluation metrics for above: CC-CCII test set, below: MDH test set.** Hollow shapes represent an expert un-aided by AI, where filled shapes represent expert with AI assistance.

## Model as expert assistant

The goal of any AI assistant model is to improve the diagnosis performance of experts. For the evaluation, first, the radiologists annotate the test set. After an appropriate amount of time, the radiologist re-annotated the set for a second time while having the diagnosis of *ai-corona* for the entire set. The test set was also shuffled the second time to eliminate any remembrance of cases. Experts' diagnosis performance is depicted in Fig 5. For the CC-CCII test set, all the experts (except the radiology Resident) had an improvement in their sensitivity. A significant improvement in the other metrics is also seen for everyone (except Senior 1). For the MDH test set, improvement in sensitivity can be seen for Senior 1 and Junior. Specificity only had an improvement in the Radiology Resident and remained unchanged for others. In every other evaluation criterion, the AI model had a positive impact on the experts' performance.

**Table 2. Evaluation results of all the studies with a 95% confidence interval using the metrics sensitivity, specificity, F1-score, Kappa, and AUC.** A "-" value indicates a lack of data. Reports in sections A, B, and C correspond to the CC-CCI test set, the MDH test set, and the MosMedData cohort, respectively.

| | Sensitivity (95% CI) | Specificity (95% CI) | F1-score (95% CI) | Kappa (95% CI) | AUC (95% CI) |
|---|---|---|---|---|---|
| **A (CC-CCII test set)** | | | | | |
| *ai-corona* | **0.972** (0.956, 0.988) | **0.968** (0.954, 0.982) | **0.970** (0.954, 0.986) | **0.935** (0.909, 0.961) | **0.997** (0.993, 0.999) |
| Zhang *et al.* [29] | 0.949 - | 0.911 - | - | - | 0.980 (0.967, 0.990) |
| Jin *et al.* [30] | 0.921 (0.918, 0.926) | 0.780 (0.770–0.789) | - | - | 0.921 (0.918, 0.926) |
| Senior 1 | 0.895 (0.874, 0.916) | 0.956 (0.945, 0.967) | 0.908 (0.892, 0.924) | 0.857 (0.837, 0.877) | - |
| Senior 1 + AI | 0.937 (0.919, 0.955) | 0.948 (0.937, 0.959) | 0.924 (0.910, 0.938) | 0.88 (0.860, 0.900) | - |
| Senior 2 | 0.909 (0.888, 0.930) | 0.917 (0.903, 0.931) | 0.884 (0.868, 0.900) | 0.816 (0.792, 0.840) | - |
| Senior 2 + AI | 0.965 (0.954, 0.976) | 0.940 (0.927, 0.953) | 0.932 (0.920, 0.944) | 0.892 (0.876, 0.908) | - |
| Junior | 0.636 (0.605, 0.667) | 0.897 (0.882, 0.912) | 0.700 (0.677, 0.723) | 0.555 (0.523, 0.587) | - |
| Junior + AI | 0.776 (0.75, 0.802) | 0.913 (0.898, 0.928) | 0.804 (0.782, 0.826) | 0.700 (0.672, 0.728) | - |
| R. Resident | 0.839 (0.813, 0.865) | 0.663 (0.639, 0.687) | 0.690 (0.670, 0.710) | 0.459 (0.426, 0.492) | - |
| R. Res. + AI | 0.853 (0.826, 0.880) | 0.778 (0.758, 0.798) | 0.760 (0.740, 0.780) | 0.599 (0.568, 0.630) | - |
| **B (MDH test set)** | | | | | |
| *ai-corona* | **0.924** (0.895, 0.953) | 0.983 (0.961, 1.000) | **0.953** (0.935, 0.971) | **0.917** (0.887, 0.947) | **0.989** (0.984, 0.994) |
| RT-PCR | 0.722 (0.661, 0.783) | - | - | - | - |
| Senior 1 | 0.857 (0.833, 0.881) | 0.979 (0.963, 0.995) | 0.903 (0.886, 0.920) | 0.858 (0.838, 0.878) | - |
| Senior 1 + AI | 0.908 (0.887, 0.929) | 0.987 (0.975, 0.999) | 0.939 (0.927, 0.951) | 0.910 (0.892, 0.928) | - |
| Senior 2 | 0.899 (0.874, 0.924) | 0.979 (0.965, 0.993) | 0.926 (0.912, 0.940) | 0.891 (0.868, 0.914) | - |
| Senior 2 + AI | 0.899 (0.877, 0.921) | 0.992 (0.983, 1.000) | 0.939 (0.928, 0.950) | 0.910 (0.894, 0.926) | - |
| Junior | 0.765 (0.738, 0.792) | 0.992 (0.982, 1.000) | 0.858 (0.838, 0.878) | 0.800 (0.775, 0.825) | - |
| Junior + AI | 0.857 (0.833, 0.881) | **1.000** (1.000, 1.000) | 0.923 (0.908, 0.938) | 0.889 (0.869, 0.909) | - |
| R. Resident | 0.882 (0.858, 0.906) | 0.92 (0.898, 0.942) | 0.864 (0.846, 0.882) | 0.794 (0.766, 0.822) | - |
| R. Res. + AI | 0.899 (0.877, 0.921) | 0.966 (0.948, 0.984) | 0.915 (0.901, 0.929) | 0.873 (0.853, 0.893) | - |
| **C (MosMedData cohort)** | | | | | |
| *ai-corona* | 0.939 (0.924, 0.954) | **0.831** (0.802, 0.860) | - | - | **0.954** (0.937, 0.971) |
| Jin *et al.* [30] | **0.945** (0.938, 0.951) | 0.661 (0.636, 0.686) | - | - | 0.933 (0.926, 0.938) |

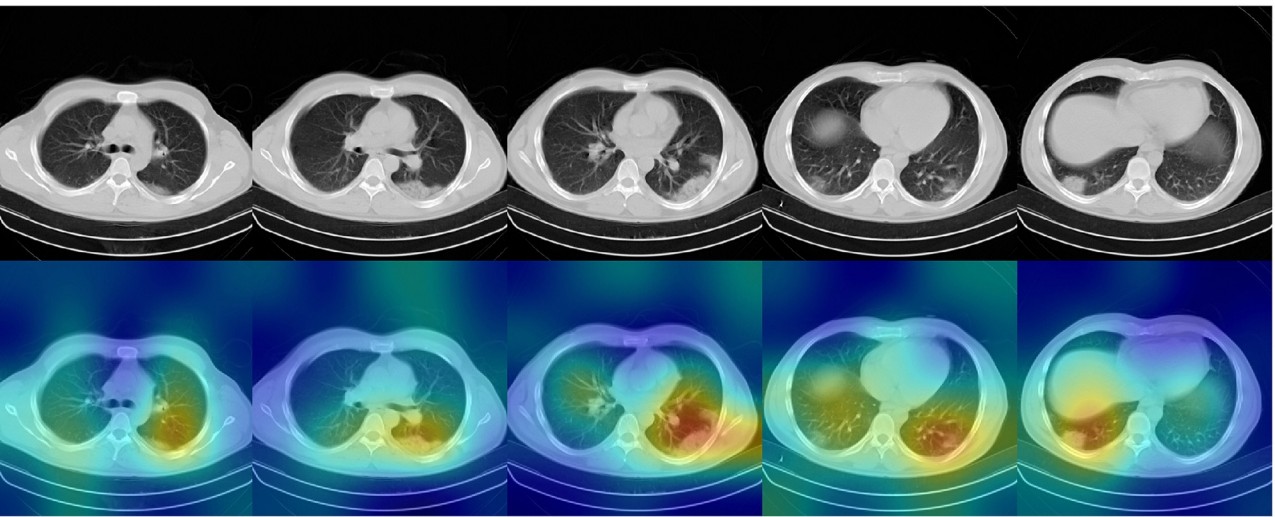

**Fig 6. Class activation maps for *ai-corona* interpretation.** This highlights which pixels in the images are important for the model's classification decision.

## Interpretation of *ai-corona*

To ensure that *ai-corona* was learning the correct imaging features, class activation maps were generated Fig 6. This is done by following the methodology described in the Introduction section. In a class activation map of a slice, more predictive areas (that hold the correct imaging features) appear brighter. Thus, the brightest areas of the class activation map correspond to regions that most influence the model's prediction.

## Additional evaluations

Additional evaluations were made as well, for which the an be found in the Supporting Information section. First, over the MDH test set, the performance of diagnosis between NCA and Normal classes was evaluated using the four metrics and was compared to the experts. Furthermore, all of the possible comparisons between every pair of classes were made to ensure the thoroughness and completeness of our evaluation which is showcased in S1–S6 Tables. As an example, this extra study showcased that radiologists perform better in diagnosing NCA from Normal compared to the AI model.

Lastly, it is important to note the speed at which different methodologies perform diagnosis. As shown in, RT-PCR is extremely slow. Moreover, our framework is faster than the best radiologist by 25 orders of magnitude. This is showcased in Table 3.

## Conclusion and discussion

We introduce *ai-corona*, a radiologist-assistant deep learning framework capable of accurate COVID-19 diagnosis in chest CT scans. Our deep learning framework was developed (training and tuning) on 5322 scans, 3985 subjects, gathered from cohorts from two countries, China

**Table 3. Diagnosis time comparison for *ai-corona* and radiologists on the 357 case test set.**

|  | *ai-corona* | Senior 1 | Senior 2 | Junior | Radiology Resident |
|---|---|---|---|---|---|
| Diagnosis Time | 12 *min.* | 360 *min.* | 300 *min.* | 320 *min.* | 400 *min.* |

and Iran, and was tested against three sets; the CC-CCII test set from China (395 scans, 252 subjects), MDH test set from Iran (357 scans, 346 subjects), and the MosMedData cohort from Russia (1110 scans, 1110 subjects). Our framework was able to learn to diagnose patients infected with COVID-19, as well as being able to distinguish between COVID-19, other types of common pneumonia (CP) such as viral and bacterial, and other non COVID-19 abnormalities (NCA). Moreover, a set of 2672 subjects was used to calculate the sensitivity of RT-PCR.

The use of multiple datasets, each with scans differing in the number of slices, and a lack of slice-specific labeling, presented a challenge for this work. To address this, we dynamically select the middle 50% of slices in each scan and feed them to a single EfficientNetB3-based feature extractor, which after an average pooling operator, will result in a single feature vector that will be classified. This method, alongside the use of only one 2D CNN, will not only make our framework more robust, but it will also make its predictions faster and capable of running on slower hardware.

Our framework was compared to two other AI models, proposed by Zhang *et al.* [29] and Jin *et al.* [30] respectively. Its diagnosis performance is also compared to that of experts and other means of diagnosis in order to achieve a comprehensive and sensible image of the framework's abilities. In the end, *ai-corona* managed to outperform the two other AI models in COVID-19 diagnosis. Our framework achieves high sensitivity, while also having a high specificity.

Our framework achieved an AUC score of 0.997 on the CC-CCII test set and performed better than the models proposed by Zhang *et al.* [29] and Jin *et al.* [30] on all four metrics. On the MDH test set, *ai-corona* gained an AUC score of 0.989 and performed mostly better in all of the metrics compared to the experts. It is worth mentioning that for our framework, diagnosing between the COVID-19 and CP classes was easier than between COVID-19 and NCA. Yet for the experts, it was the opposite. RT-PCR, as another method of diagnosis, had a sensitivity of 0.722 at best, worse than all the experts and the AI. At last, our framework gained a 0.954 AUC score on the MosMedData cohort, which outperforms Jin *et al.* [30]. A complete report of these evaluations can be found in Fig 3 through Fig 5 and Table 2.

In COVID-19 diagnosis, *ai-corona*'s impact on assisting experts' diagnosis was evaluated, which in COVID-19 diagnosis, mostly indicates a positive improvement on at least their sensitivity or specificity. This improvement is most noticeable for the Junior radiologist and the radiology Resident. Additionally, incorporation of the class activation maps in the experts' diagnosis can help them examine the involved regions better.

On having a positive impact on experts' diagnosis, two cases are discussed here to showcase how *ai-corona* made experts change their minds for good in suspicious cases. At least one expert misdiagnosed Fig 7(a)'s case as NCA at first, but upon seeing the AI's diagnosis, correctly diagnosed as COVID-19. This expert cited seeing Peribrochovascular distribution, which is not common in COVID-19 (no subpleural distribution), as the reason for their misdiagnosis. In addition, Fig 7(b)'s case was initially misdiagnosed as COVID-19 by at least one expert, but was changed correctly to NCA when seeing the AI's correct diagnosis. They cited that cavity, centrilobular nodule, mass, and mass-like consolidations are not commonly seen in COVID-19 pneumonia and might implicate other diagnostics.Fig 3On the other hand, the existence of error in CT-based diagnosis, both for *ai-corona* and experts, encourages us to study the cause for such errors, which might lead to better and more accurate predictions, or point out any if existing fundamental flaws in CT-based diagnosis. Fig 7(a)'s case was misdiagnosed as COVID-19 by all the experts. Our framework, while correctly diagnosing for NCA, was not able to change the experts' minds. In a consensual final report, the experts cite that Mediastinal and bilateral hilar adenopathies were seen, as well as Anterior mediastinal soft-tissue density. In addition, Diffuse bilateral interstitial infiltrations were detected with crazy

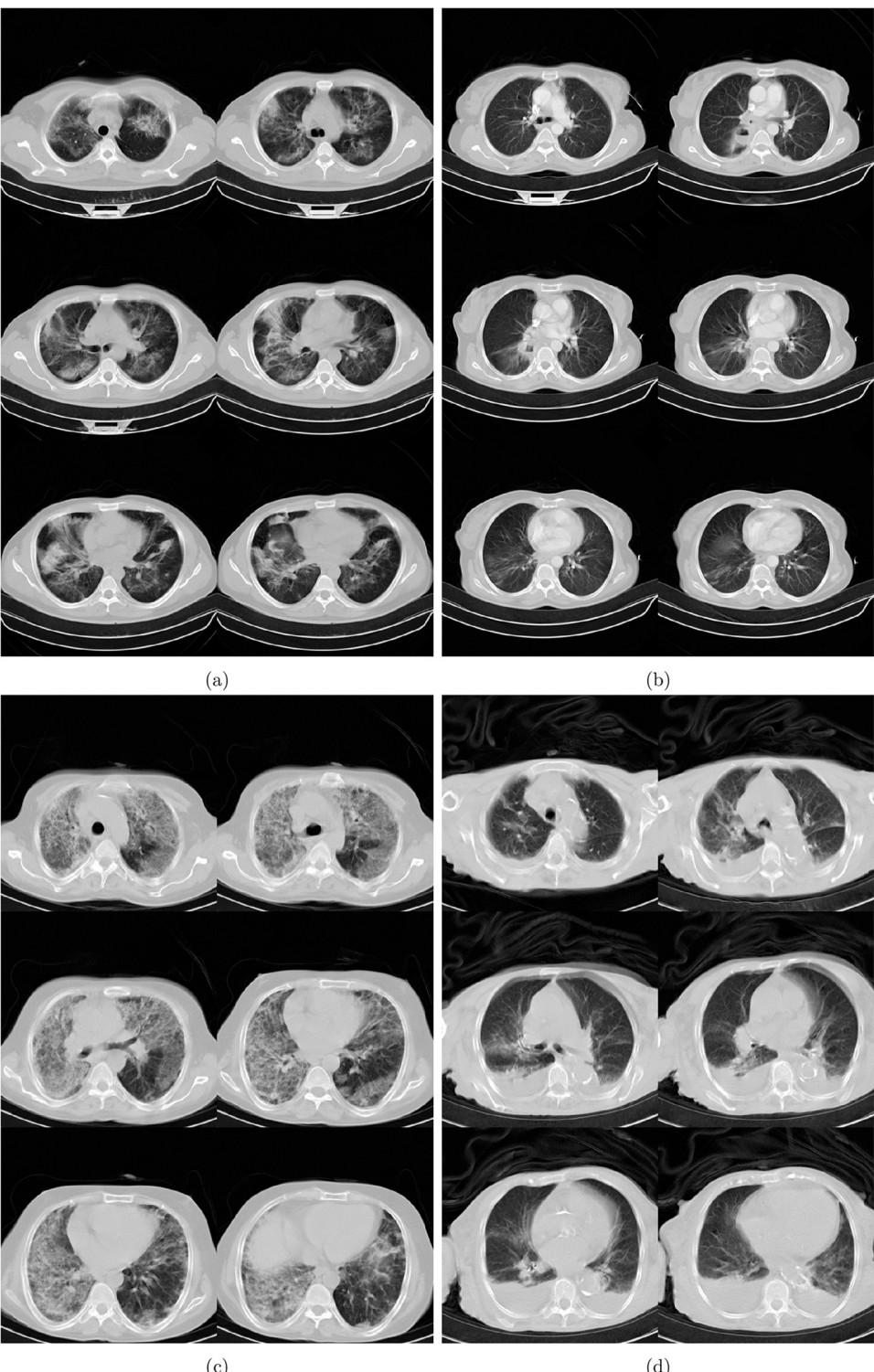

(a)

(b)

(c)

(d)

**Fig 7.** (a), (b), and (c), are the chest CT scans of patients who were initially misdiagnosed by at least one radiologist but were then diagnosed correctly upon incorporating ***ai-corona* 's correct prediction.** (d) shows the chest CT scans of a patient that was misdiagnosed by *ai-corona* and radiologists.

paving pattern, ground glass, and traction bronchiectasis, mainly in the right lung and partial volume loss of the right lung. Also, the position of central venous catheter tip was seen in the left brachiocephalic vein.

The success of AI in medical imaging-based diagnosis has been proven by this work and many others before it. *ai-corona* can positively influence an expert's opinion and improve the speed at which the subject screening process occurs, such that it helps critical cases get the care they urgently need faster.

But our work has its own drawbacks and shortcomings. Since the gathering of a dataset with better labeling (one that alongside its accurate annotations also accompanies localization and slice labels) is time and resource consuming, we decided to opt for an approach that favors robustness and is capable of learning on a simpler dataset. Developing our framework on a better dataset would certainly improve its performance. In addition, the CP class contains all kinds of conditions and diseases that cause pneumonia. As each of these conditions and diseases have their own distinct imaging features, having separate classes for them, especially Influenza-A, would improve the framework's performance. Lastly, our framework's learning would certainly benefit from more cases that are positive for COVID-19, yet have a negative RT-PCR result. As these cases are mostly experiencing the early stages of the infection, diagnosing them is more difficult. Moreover, classifying cases with a negative RT-PCR as non COVID-19 is illogical and their labeling protocol should be something else.

In the future, approaches that do a better job incorporating clinical reports with the imaging data should be explored. In conclusion, with the individual drawbacks of diagnosing based on clinical representation, RT-PCR, and CT-based diagnosis, a method comprised of all three would definitely yield the most accurate diagnosis of COVID-19.

## Supporting information

**S1 Fig. The cascade structure of the MDH. The number of subjects and scans in each split and set is indicated.** The preliminary dataset was cleaned, by removing abdomen and high-resolution CT scans. The train and tuning sets were labeled by two expert radiologists. The NCA and Normal classes of the test set was re-annotated by three expert radiologist (one new). The COVID-19 class are patients that meet our criteria and were hospitalized for more than three days.
(TIF)

**S2 Fig. The left panel corresponds to the distribution of image slices for cases in the MDH, the middle panel shows the distribution of age, while the right panel illustrates the sex distribution of cases in the MDH.**
(TIF)

**S3 Fig. The ROC diagram representing the performance of various pipelines for the different combinations of comparison.** The Solid black line is for *ai-corona* by adapting different discrimination threshold value which is used to convert the continuous probability to binary "Yes" or "No" results. The filled triangle symbols are the (1-specificity, sensitivity) for the individual clinical experts, while the filled circle symbols are for the model-assisted radiologist. The inset plots magnify the highest part of sensitivity and specificity.
(TIF)

**S1 Table. The quantitative evaluation of *ai-corona*, radiologists, and AI-assisted radiologists' performance results for differentiating between the COVID-19 class and the Normal class at a 95% confidence interval.**
(PDF)

**S2 Table. The quantitative evaluation of *ai-corona*, radiologists, and AI-assisted radiologists' performance results for differentiating between the COVID-19 class and the NCA class at a 95% confidence interval.**
(PDF)

**S3 Table. The quantitative evaluation of *ai-corona*, radiologists, and AI-assisted radiologists' performance results for differentiating between the NCA class and the other classes at a 95% confidence interval.**
(PDF)

**S4 Table. The quantitative evaluation of *ai-corona*, radiologists, and AI-assisted radiologists' performance results for differentiating between the Normal class and the other classes at a 95% confidence interval.**
(PDF)

**S5 Table. The quantitative evaluation of *ai-corona*, radiologists, and AI-assisted radiologists' performance results for differentiating between the NCA class and the Normal class at a 95% confidence interval.**
(PDF)

**S6 Table. The quantitative evaluation of *ai-corona*, radiologists, and AI-assisted radiologists' performance results for differentiating between the NCA class and the Normal class at a 95% confidence interval.**
(PDF)

## Acknowledgments

Our framework is available to expert professionals and the public healthcare via the website at *ai-corona*.com for free and unlimited use, where they can upload a chest CT scan and have it diagnosed for COVID-19 infection. The authors would like to express their gratitude to the Masih Daneshvari Hospital, Zahra Yousefi, Abbas Danesh, Negar Bandegani, and Shahram Kahkouee for all their hard work and assistance in this project. We appreciate Prof. Babak A. Ardekani at the Nathan S. Kline Institute and Erfan Zabeh, a Ph.D student at Columbia University, for their excellent comments and editing the manuscript. The computational part of this work was carried out on the Brain Engineering Research Center and the High-Performance Computing Cluster of the Institute for Research in Fundamental Sciences (IPM).

## Author Contributions

**Conceptualization:** Mehdi Yousefzadeh, Parsa Esfahanian, Atefeh Abedini, Reza Lashgari.

**Data curation:** Mehdi Yousefzadeh, Atefeh Abedini, Seyed Alireza Nadji, Sara Haseli, Mehrdad Bakhshayesh Karam, Arda Kiani, Jafar Roshandel.

**Formal analysis:** Mehdi Yousefzadeh, Parsa Esfahanian, Seyed Mohammad Sadegh Movahed, Saeid Gorgin, Dara Rahmati, Reza Lashgari.

**Investigation:** Mehdi Yousefzadeh, Parsa Esfahanian, Atefeh Abedini, Reza Lashgari.

**Methodology:** Mehdi Yousefzadeh, Parsa Esfahanian, Seyed Mohammad Sadegh Movahed, Saeid Gorgin, Dara Rahmati, Atefeh Abedini, Reza Lashgari.

**Project administration:** Reza Lashgari.

**Resources:** Atefeh Abedini, Seyed Alireza Nadji, Sara Haseli, Mehrdad Bakhshayesh Karam, Arda Kiani, Jafar Roshandel, Reza Lashgari.

**Software:** Mehdi Yousefzadeh, Parsa Esfahanian, Reza Lashgari.

**Supervision:** Atefeh Abedini, Reza Lashgari.

**Validation:** Seyed Alireza Nadji, Sara Haseli, Mehrdad Bakhshayesh Karam, Arda Kiani, Meisam Hoseinyazdi, Jafar Roshandel, Reza Lashgari.

**Visualization:** Mehdi Yousefzadeh, Parsa Esfahanian, Reza Lashgari.

**Writing – original draft:** Mehdi Yousefzadeh, Parsa Esfahanian, Seyed Mohammad Sadegh Movahed, Saeid Gorgin, Dara Rahmati, Atefeh Abedini, Reza Lashgari.

**Writing – review & editing:** Mehdi Yousefzadeh, Parsa Esfahanian, Atefeh Abedini, Reza Lashgari.

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
