## [Decision Letter · Decision Letter 0]

5 Mar 2021

PONE-D-21-04656

ai-corona: Deep Radiologist-Assistant for COVID-19 Diagnosis in Chest CT Scans

PLOS ONE

Dear Dr. Lashgari,

Thank you for submitting your manuscript to PLOS ONE. After careful consideration, we feel that it has merit but does not fully meet PLOS ONE’s publication criteria as it currently stands. Therefore, we invite you to submit a revised version of the manuscript that addresses the points raised during the review process.

The manuscript had be reviewed by 2 reviewers and both reviewers were of the view that manuscript describes a technically sound piece of scientific research and recommended minor revision. However, they had made certain observations to improve your work.

After thorough consideration of comments of reviewers, my decision is "minor revision". Please incorporate comments raised by both reviewers. 

**Additional Note to Authors:** I noted that one of the reviewers has asked for more context in the literature review, and suggested specific papers to be cited. While you may take on-board their suggested papers if you feel that they are relevant for your manuscript, or just take on-board the general suggestion for providing some more context in the literature review, there is no requirement from the journal to cite these papers

We look forward to receiving your revised manuscript.

Kind regards,

Gulistan Raja

Academic Editor

PLOS ONE

Journal Requirements:

5. Please ensure that you refer to Figures 4, 8, 9 and 10 in your text as, if accepted, production will need this reference to link the reader to each figure.

6. We note you have included tables to which you do not refer in the text of your manuscript. Please ensure that you refer to Tables 3-8 in your text; if accepted, production will need this reference to link the reader to each Table.

Reviewers' comments:

Reviewer's Responses to Questions

**Comments to the Author**

1. Is the manuscript technically sound, and do the data support the conclusions?

Reviewer #1: Yes

Reviewer #2: Yes

2. Has the statistical analysis been performed appropriately and rigorously? 

Reviewer #1: Yes

Reviewer #2: N/A

3. Have the authors made all data underlying the findings in their manuscript fully available?

Reviewer #1: Yes

Reviewer #2: Yes

4. Is the manuscript presented in an intelligible fashion and written in standard English?

Reviewer #1: No

Reviewer #2: Yes

5. Review Comments to the Author

Reviewer #1: The paper describes a CNN-based model for classification of 3D-CTs as either presenting COVID-19, having several other abnormalities, or being normal. An ImageNet pretrained EfficientNetB3 is used as a slice-wise feature extractor. Then, the features from all slices are combined by averaging and classified using a fully connected layer. The model achieves compelling results, outperforming on its own even experts helped by the model for the task of COVID-19 vs others classification. The datasets employed and the validation techniques are all sound and there is a very complete array of numerical comparisons with 4 radiologists. I recommend it for publication, although there are some comments that should be addressed:

The paper requires English review. I am no native English speaker, but only in the abstract, I can find already several grammatical mistakes: “We employed three independent dataset”, “Our results show thatai-coronaoutperforms all”, “for which it gained AUC score of 0.997”, “our framework’s diagnosis capabilities was evaluated”.

Line 77: The differences between NCA (non COVID-19 abnormal), non-pneumonia, and normal classes should be clarified, and also what pathologies might represent each of the first two.

Line 176: Average pooling is employed to combine the final activation maps from each slice into a single activation map for the whole CT. However, this ignores any kind of spatial information contained along the z-axis, as it is all averaged. Furthermore, using average pooling might bury any positive detection among many negative ones. Have you tried using another combination method that is aware of the z-axis information (such as 3D pooling, instead of 2D, or using 3D convolutions)? Have you tried using max-pooling instead of average pooling? Also, I would have suggested to add the previous and next slice to the input of each EfficientNet feature extractor, instead of repeating the same slice three times, as way to add some z-axis information to each prediction.

Line 189: Do you use EfficientNetB5 or B3? B5 is mentioned here, but B3 is mentioned in the rest of the text. Also, Figure 1 shows that the input images are of resolution 512x512, but that is not the input resolution of the EfficientNetB3, nor the B5.

Reviewer #2: Minor Revision

# In the introductory section, the first paragraph should discuss about COVID-19 pandemic, number cases, deaths, recovered and the taxonomy of the virus etc.

# Author should stress on why their contribution stand out or achieved better result than others.

# There should be bullet points on authors contribution at the end of introduction.

# Author should add a paragraph on COVID-19 vaccines.

# Author need to add a section on “Related work” and a table to summarized contributions in terms of dataset and performance evaluation.

# Author need to add a limitations and future work.

# Authors can also add these references

1. Chowdhury, M. E., Rahman, T., Khandakar, A., Mazhar, R., Kadir, M. A., Mahbub, Z. B., ... & Islam, M. T. (2020). Can AI help in screening viral and COVID-19 pneumonia?. IEEE Access, 8, 132665-132676.

2. Ibrahim, A. U., Ozsoz, M., Serte, S., Al-Turjman, F., & Yakoi, P. S. (2021). Pneumonia classification using deep learning from chest X-ray images during COVID-19. Cognitive Computation, 1-13.

3. Burki, T. K. (2020). The Russian vaccine for COVID-19. The Lancet Respiratory Medicine, 8(11), e85-e86.

6. PLOS authors have the option to publish the peer review history of their article (what does this mean?). If published, this will include your full peer review and any attached files.

Reviewer #1: **Yes: **Oscar José Pellicer Valero

Reviewer #2: **Yes: **Abdullahi Umar Ibrahim PhD

---

## [Author Response · Author response to Decision Letter 0]

15 Apr 2021

Dear Dr. Gulistan Raja,

Academic Editor

PLOS ONE

Thank you for giving us the opportunity to resubmit our revised manuscript “ai-corona: Deep Radiologist-Assistant for COVID-19 Diagnosis in Chest CT Scans”. Ms. No: PONE-D-21-04656'' to PLOS ONE. In this version of the manuscript, we have addressed all the criticisms of the reviewers and made changes throughout the main manuscript text. We appreciate the time and effort that you and the reviewers have dedicated to providing your valuable feedback on our manuscript. We are grateful for the reviewer’s careful reading of the paper and their excellent criticisms and recommendations to improve the paper. The criticisms had led to significant improvements made in the manuscript. We also revised the title of the manuscript as follows: “ai-corona: Radiologist-Assistant Deep Learning Framework for COVID-19 Diagnosis in Chest CT Scans”. Below, we detail how we have responded to the two reviewers’ criticisms. 

Sincerely,

Reza Lashgari, Ph.D 

Summary of the changes: 

All changes in the manuscript were highlighted in blue color.

• The manuscript text was justified.

• The entire manuscript has been carefully edited by a native English speaker to improve grammar and readability.

• References, numbering, and the references section were reviewed and double-checked.

• We incorporated the data availability section into the final manuscript.

• Zero padding to 3 decimal points for coherence was added in Table 3. To prevent visual congestion on the "supporting information" version, blue highlighting was left out.

• Changed the positioning of some of the tables and figures for better visual presentation.

• We added the author contributions in the end of manuscript.

Manuscript Title:

We revised the manuscript tile as “ai-corona: Radiologist-Assistant Deep Learning Framework for COVID-19 Diagnosis in Chest CT Scans”. It is highly appreciated if the respected editor and reviewers accept the revised title as we changed in this revision. 

Changes to the Affiliations:

 The affiliations were revised and the symbol was removed from the authors D. Rahmati and A. Kiani.

Changes to the Abstract: 

We significantly revised the abstract according to the reviewer’s comment. The new abstract follows the same semantic and sentencing structure as the previous one but written in more eloquent English letting a broader range of audiences understand the science behind our article.

Changes to the Introduction Section:

• A sentence to discuss the COVID-19 pandemic, number of cases, deaths, and etc. was added to the first paragraph with appropriate citations.

• Added a sentence to discuss COVID-19 vaccines and the challenges with appropriate citation.

• The Paper's contributions and achievements were summarized at the end of the introduction section.

• The manuscript has been carefully edited by a native English speaker to improve grammar and eloquence.

Changes to the Materials and Methods Section:

• Added reference to the figures.

• Addressed the individual reviewer points.

• The entire manuscript has been carefully edited by a native English speaker to improve grammar and readability.

Changes to the Results Section:

• We included references to the figures and tables. 

• All of the figures and tables in the Supporting Information section are now referenced.

• Added a diagnosis time comparison table and its appropriate referencing.

• The paper has been carefully edited by a native English speaker to improve grammar and readability.

Changes to the Conclusions Section:

The conclusion section has been carefully edited by a native English speaker to improve grammar and readability.

Changes to the Supporting Information Section:

• Slight changes were made to the Supporting Fig 1.

• Captions for figures and tables were modified.

• Multiple grammatical corrections were made.

Response to reviewers comments:

We sincerely appreciate all your valuable comments and suggestions, which helped us to improve the quality of the article. Our responses to the respected reviewer’s comments are described below in a point-to-point manner. The changes, suggested by the reviewers, have been described to the new version of the manuscript (highlighted in blue color within the document).

Response to Reviewer #1:

The paper describes a CNN-based model for classification of 3D-CTs as either presenting COVID-19, having several other abnormalities, or being normal. An ImageNet pretrained EfficientNetB3 is used as a slice-wise feature extractor. Then, the features from all slices are combined by averaging and classified using a fully connected layer. The model achieves compelling results, outperforming on its own even experts helped by the model for the task of COVID-19 vs others classification. The datasets employed and the validation techniques are all sound and there is a very complete array of numerical comparisons with 4 radiologists. I recommend it for publication, although there are some comments that should be addressed:

 Thank you for the excellent comments and constructive review.

The paper requires English review. I am no native English speaker, but only in the abstract, I can find already several grammatical mistakes: “We employed three independent dataset”, “Our results show thatai-coronaoutperforms all”, “for which it gained AUC score of 0.997”, “our framework’s diagnosis capabilities was evaluated”.

Thank you. The manuscript has been carefully edited by a native speaker. 

Line 77: The differences between NCA (non COVID-19 abnormal), non-pneumonia, and normal classes should be clarified, and also what pathologies might represent each of the first two.

 We thank the reviewer for their feedback. We agree with the reviewer’s comment regarding the lack of clarity in the classification of covid categories and as a result, we have elaborated upon these differences more carefully. For the difference between the NCA and the non-pneumonia classes, we added the following explanation to the Data subsection: "having the NCA class is crucial, as it includes abnormalities such as atelectasis, cardiomegaly, lung emphysematous, hydropneumothorax, pneumothorax, cardiopulmonary edema, cavity, fibrocavitary changes, fibrobronchiectatic, mass, and nodule."

Line 176: Average pooling is employed to combine the final activation maps from each slice into a single activation map for the whole CT. However, this ignores any kind of spatial information contained along the z-axis, as it is all averaged. Furthermore, using average pooling might bury any positive detection among many negative ones. Have you tried using another combination method that is aware of the z-axis information (such as 3D pooling, instead of 2D, or using 3D convolutions)? 

Thank you. The reviewer’s concern about the neglection of z-axis information is valid. We also were aware of this reduction and accordingly we compared our algorithm with other combination methods. In the initial phases of our model development prototyping, we experimented with 3D-CNN models that extract the z-axis information such as 3D-ResNet. The results were atrocious and this method was discarded.

Have you tried using max-pooling instead of average pooling? 

We initially used max-pooling in our model's last layer (following the same as the MRNet paper). But our experiments indicate that average pooling has better results comparing to max-pooling in our algorithm so we only reported the algorithm with average-pooling. As we know the Average-pooling encourages the network to identify the complete extent of the object, whereas max-pooling restricts that to only the very important features, and might miss out on some details. Based on this intuition we speculate the class information is encoded in population-pixel level so the average-pooling performs better in our experiments.

Also, I would have suggested to add the previous and next slice to the input of each EfficientNet feature extractor, instead of repeating the same slice three times, as way to add some z-axis information to each prediction.

Thank you for the suggestion. For the model input size, we experimented with two settings; EfficientNet's standard input size and the scan's default size of 512*512. The second one was shown significant results and better descriptions.

It is worth mentioning that our model's input is a 3-channel grayscale image that is the standard practice of many experts in the literature. As we were satisfied with our results and we didn't have time to experiment with other methods, we decided to commit to this practice.

Line 189: Do you use EfficientNetB5 or B3? B5 is mentioned here, but B3 is mentioned in the rest of the text. Also, Figure 1 shows that the input images are of resolution 512x512, but that is not the input resolution of the EfficientNetB3, nor the B5.

Thank you for the important point and sorry for the error typo. The employed feature extractor is based on EfficientNetB3, but was mistakenly typed as EfficientNetB5. This is fixed now.

Response to Reviewer #2:

Thank you for raising the following points and constructive review and comments:

# In the introductory section, the first paragraph should discuss about COVID-19 pandemic, number of cases, deaths, recovered and the taxonomy of the virus etc.

You have raised an important point here and have incorporated your suggestion throughout the manuscript. Accordingly, we briefly discussed the COVID-19 pandemic, number of cases, deaths, and vaccination in the first paragraph of the introduction. Please see the highlighted blue color in the main text of the revised manuscript. 

# Author should stress on why their contribution stand out or achieved better result than others.

In all the tables, the best results were made bold and we also added a paragraph at the end of the introduction.

# There should be bullet points on authors contribution at the end of introduction.

Thank you. We reimplemented the author’s contribution section as you mentioned.

# Author should add a paragraph on COVID-19 vaccines.

Thanks for raising this point. Following the reviewer’s suggestion and the global attempt in the development of the COVID-19 vaccines and their effect on the treatment of newly infected subjects, we briefly added a sentence in the first paragraph of the introduction section elaborating upon this issue. Particularly with the low protection capabilities of existing vaccines, we claim the development of radiologist assistance technologies for detection and treatment of COVID-19 are still essential. 

# Author need to add a section on “Related work” and a table to summarized contributions in terms of dataset and performance evaluation.

Thank you. We also believe the contribution of each data set should be reported individually, accordingly to the introduction section, the related works were sufficiently discussed. Also, the results for the two similar works were present in Table 2. 

# Author need to add a limitations and future work.

Our work's limitations and future directions were described in the conclusion and discussion. 

 # Authors can also add these references

1. Chowdhury, M. E., Rahman, T., Khandakar, A., Mazhar, R., Kadir, M. A., Mahbub, Z. B., ... & Islam, M. T. (2020). Can AI help in screening viral and COVID-19 pneumonia?. IEEE Access, 8, 132665-132676.

2. Ibrahim, A. U., Ozsoz, M., Serte, S., Al-Turjman, F., & Yakoi, P. S. (2021). Pneumonia classification using deep learning from chest X-ray images during COVID-19. Cognitive Computation, 1-13.

3. Burki, T. K. (2020). The Russian vaccine for COVID-19. The Lancet Respiratory Medicine, 8(11), e85-e86.

Thank you. We included these two references (1 and 2) in the introduction and reference sections.

---

## [Editor Report · Decision Letter 1]

19 Apr 2021

ai-corona: Radiologist-Assistant Deep Learning Framework for COVID-19 Diagnosis in Chest CT Scans

PONE-D-21-04656R1

Dear Dr. Lashgari,

We’re pleased to inform you that your manuscript has been judged scientifically suitable for publication and will be formally accepted for publication once it meets all outstanding technical requirements.

Kind regards,

Gulistan Raja

Academic Editor

PLOS ONE
---

## [Editor Report · Acceptance letter]

27 Apr 2021

PONE-D-21-04656R1 

*ai-corona*: Radiologist-Assistant Deep Learning Framework for COVID-19 Diagnosis in Chest CT Scans 

Dear Dr. Lashgari:

I'm pleased to inform you that your manuscript has been deemed suitable for publication in PLOS ONE. Congratulations! Your manuscript is now with our production department. 

Kind regards, 

on behalf of

Dr. Gulistan Raja 

Academic Editor

PLOS ONE